# Long-term outcomes of combination therapy with stereotactic body radiation therapy plus cryoablation using liquid nitrogen for stage I non-small cell lung cancer with tumors ≥2 cm

Hiroaki Nomori[1]*, Cong Yue[2], Hiroyoshi Iguchi[3], Kenichi Kashihara[3], Ryuichi Wada[4], Tsutomu Saito[5]

1 Department of Thoracic Surgery, Kashiwa Kousei General Hospital, Kashiwa, Chiba, Japan,
2 Department of Thoracic Surgery, Tokyo University Hospital, Bunkyo-ku, Tokyo, Japan, 3 Department of Radiology, Kashiwa Kousei General Hospital, Kashiwa, Chiba, Japan, 4 Department of Pathology, Kashiwa Kousei General Hospital, Kashiwa, Chiba, Japan, 5 Department of Radiology, Sonodakai Radiation Clinic, Adachi-ku, Tokyo, Japan

* hnomori@qk9.so-net.ne.jp

## Abstract

### Objectives

Radiation and ablation therapy are used to treat stage I non-small cell lung cancer. However, the local control and survival rates remain unsatisfactory for tumors ≥2 cm. Therefore, this study explored the effectiveness of combination therapy of stereotactic body radiation plus cryoablation in this patient population.

### Methods

This retrospective observational study included patients with stage I non-small cell lung cancer ≥2 cm who underwent stereotactic body radiation therapy followed by cryoablation between 2015 and 2023. All tumors were pathologically diagnosed before treatment. Patients received stereotactic body radiation therapy (40–60 Gy/4–10 fractions, median biological effective dose: 100 Gy), followed by cryoablation using liquid nitrogen within 3 weeks of radiation therapy. Study outcomes included local control and survival rates, as well as adverse events.

### Results

Sixty-four patients were included in the study. The mean tumor diameter was 2.7±0.5 (range, 2.0–4.0) cm. The median follow-up duration was 74 (range, 3–111) months. Five patients (8%) experienced local recurrence after treatment (range: 8–61 months). The 5-year local control and overall survival rates were 93% and 74%, respectively. No patient experienced treatment-related mortality. The most frequent complication was post-cryoablation pneumothorax (40%), with a median drainage

**Data availability statement:** Anonymized data are all contained in Supporting information files.

**Funding:** The author(s) received no specific funding for this work.

**Competing interests:** No authors have competing interests.

**Abbreviations:** BED, Biological effective dose; CI, 95% Confidence interval; CTCAE, Common terminology criteria for adverse events; DSS, Disease-specific survival; FDG-PET, Fluorodeoxyglucose-positron emission tomography; FEV1, Forced expiratory volume in 1 s; IQR, Interquartile range; LC, Local control; NSCLC, Non-small cell lung cancer; OS, Overall survival; RFA, Radiofrequency ablation; RFS, Recurrence-free survival; ROC, Receiver-operating characteristic; RTOG, Radiation Therapy Oncology Group; SBRT, Stereotactic body radiation therapy; SUV, Standardized uptake value; TNM, Tumor, Node, Metastasis.

period of 2 days. Five patients (8%) experienced radiation pneumonitis; four patients had grade 2 severity and one had grade 3 severity, but all these patients recovered without sequelae.

## Conclusion

Combination therapy with stereotactic body radiation therapy followed by cryoablation is a feasible and favorable treatment for stage I non-small cell lung cancer with tumors ≥2 cm.

## Introduction

Stereotactic body radiotherapy (SBRT) is the first choice of treatment for inoperable patients with stage I non-small cell lung cancer (NSCLC). However, the overall survival (OS) rate remains significantly lower than the local control (LC) rate (40–83% vs. ≥90%) [1–8]. A meta-analysis reported a significantly shorter OS rate after SBRT compared to that after surgery in patients with stage I NSCLC [9]. The short OS duration after SBRT may be attributed to the high comorbidity in inoperable patients, but it could also result from residual tumors after SBRT. The residual tumors are usually difficult to detect on computed tomography (CT) due to large scars that develop after SBRT and may metastasize, decreasing the OS rate. Additionally, most studies on SBRT have reported a 3-year OS rate [1–4], which is too short to adequately assess treatment outcomes in stage I NSCLC. The Radiation Therapy Oncology Group (RTOG) 0618 trial reported a 4-year OS rate of 56% after SBRT in 26 patients with T1–T2 N0M0 NSCLC [5], and the RTOG 0236 study reported a 5-year OS rate of 40% in 55 patients with stage I NSCLC [6]. Furthermore, Shintani et al. reported that local recurrence occurred more than 5 years after SBRT in 7 of 216 patients (3%) with NSCLC [10]. Together, these reports suggest that the 3 years is insufficient for assessing OS after SBRT.

Ablation therapies, such as cryoablation or radiofrequency ablation (RFA), are alternative therapies for treating stage I NSCLC [11–14]. The 3-year LC and OS rates after cryoablation for stage I NSCLC are comparable at 90% and 95%, respectively, unlike after SBRT [11]. However, the LC rates after both cryoablation and RFA are lower for NSCLCs with tumors ≥2 cm than those with tumors <2 cm [11,12]; this is also true for SBRT results [9]. Therefore, both ablation therapies and SBRT are limited based on a tumor size ≥2 cm.

Tumor control mechanisms differ for SBRT, cryoablation, and RFA. SBRT injuries the tumor DNA, whereas cryoablation induces the formation of ice crystals, leading to cell membrane destruction and subsequent intracellular dehydration. In contrast, RFA leads to coagulation necrosis due to high temperatures. This mechanistic variability could be advantageous, when using combination of these therapies, which may improve LC and OS outcomes for stage I NSCLC with tumors ≥2 cm, but combination therapies have not been explored.

RFA increases the incidence of pneumonitis in patients with NSCLC who have previously undergone radiotherapy [13], but cryoablation does not affect the incidence of

pneumonitis between patients with and without interstitial pneumonitis [14]. Therefore, we selected cryoablation instead of RFA for combination with SBRT.

Regarding the combination therapy with SBRT plus cryoablation, SBRT should be performed first, as cryoablation decreases tumor vascularity, which in turn decreases tumor radiosensitivity [15]. Additionally, scarring around the tumor occurs immediately after cryoablation, which impedes SBRT planning. In contrast, SBRT usually does not induce inflammation within one month of treatment. Therefore, we performed SBRT first, followed by cryoablation 2–3 weeks after the SBRT in patients with stage I NSCLC with tumors ≥2 cm, which assessed the long-term effectiveness with a median follow-up period of 74 months.

## Materials and methods

### Study design

This descriptive retrospective cohort study adhered to the Strengthening the Reporting of Observational Studies in Epidemiology guidelines [16] and was conducted in accordance with the Declaration of Helsinki. The two ethics committees of the participating institutions approved this study in 2015 (approval no. 15−011) and 2020 (approval no. 20−03).

### Eligibility

Patients with tumors pathologically diagnosed as NSCLC before treatment via needle or bronchoscope biopsy were eligible. The tumor stage was confirmed using 18F-fluorodeoxyglucose-positron emission tomography (FDG-PET). The inclusion criteria were those (1) with stage I peripheral-type NSCLC with a tumor size ≥2 cm based on the 8th edition of the Tumor, Node, Metastasis (TNM) classification by the International Association for the Study of Lung Cancer [17]; (2) with an Eastern Cooperative Oncology Group performance status of 0–2; (3) ineligible for surgery owing to comorbidities, low forced expiratory volume in 1 s ($FEV_1$), and age (≥80 years); and (4) who preferred the combined SBRT/cryoablation treatment over surgery. The exclusion criteria were those (1) with active interstitial pneumonitis or who underwent a previous treatment for interstitial pneumonitis; and (2) with tumors located outside the reach of the cryoprobe, such as near the hilar pulmonary artery or vein, aorta, superior vena cava, heart, and esophagus. All patients were informed that the standard therapy for stage I NSCLC was surgery and not ablation therapies or SBRT, as these had lower LC rates compared to surgery. All patients provided written informed consent after receiving this information and hearing the potential risks and benefits of the combined SBRT/cryoablation treatment.

### SBRT

SBRT was administered using a linear accelerator (Elekta Versa HD; Elekta Co. Ltd., Tokyo, Japan). Respiratory motion was managed using three sets of conventional free-breathing CT simulation images, with an abdominal compression device for breath-hold (Pressure Belt; Euro Meditech Co., Ltd., Tokyo, Japan).

The internal tumor volume, encompassing the full range of tumor motion, was delineated using the second and third image sets and referenced to the clinical target volume defined in the first image set. A planning target volume margin of 5 mm was added to the internal tumor volume based on three RTOG studies (0618, 0236, and 0915 [5,6,8]).

Radiation oncologists determined the fractionation to optimize the planning target volume coverage while adhering to the dose constraints for the organs at risk based on the RTOG protocols. Each fractionation schedule delivered a biological effective dose (BED) of 100 Gy ($\alpha/\beta = 10$ Gy), except to patients ≥80 years old or those with interstitial pneumonitis on CT image, where it was lowered to a minimum of 80 Gy.

Treatments were administered once daily on consecutive days. Before each irradiation session, a flat-panel cone-beam CT scan was performed to identify the tumor location, which was compared to the clinical target volume from the treatment plan. Positional discrepancies were corrected manually or using a six-degree-of-freedom treatment couch.

## Cryoablation

Cryoablation using liquid nitrogen (IceSense 3; IceCure Medical Ltd., Caesarea, Israel) was performed 2–3 weeks after SBRT using a 3.4-mm diameter cryoprobe (**Fig 1**) [11]. A guide needle kit (Daimon coaxial system; Silux Co., Kawaguchi, Japan), comprising a 21-G guide needle and an 11-G stainless-steel coaxial system with inner and outer sheaths, was used to guide the cryoprobe to the tumor. This device is necessary, as directly advancing the cryoprobe to the tumor is difficult under spontaneous breathing.

The patient was placed in the supine or prone position on the CT table, depending on the tumor's location. Sedatives, such as pethidine hydrochloride (35 mg) and midazolam (2–3 mg), were administered. Following local anesthesia, the guide needle was introduced into the tumor under CT guidance (**Fig 2**). The inner and outer sheaths were advanced over the guide needle to reach the tumor. Once the outer sheath reached the target position, the guide needle and inner sheath were removed. The cryoprobe was then introduced into the outer sheath, and freezing was initiated.

Cryoablation was typically performed using a single cryoprobe and three freeze-thaw cycles with the following sequence: 5-min freeze, 8-min passive thaw; 8-min freeze, 10-min passive thaw; and 8-min freeze, 4-min active thaw (total treatment duration: 43 min). Two cryoprobes were used for tumors ≥3 cm, each-connected to a separate IceSense3 device, as one device is required per cryoprobe.

## Follow-up

Follow-up CT scans were conducted every 3 or 4 months until 3 years after treatment, after which patients were followed up in intervals of at least 6 months until the final follow-up day. The LC and survival durations were calculated from the first day of SBRT. The cause of death was classified as either death due to the primary disease or death due to other diseases. Follow-up data were collected from chart abstractions on September 5, 2024.

## Safety assessments

Posttreatment adverse events were graded based on the Common Terminology Criteria for Adverse Events (CTCAE) version 3.0 [18]. Pneumonitis was diagnosed based on CT findings and symptoms. Pneumothorax after cryoablation was defined as cases requiring thoracic drainage.

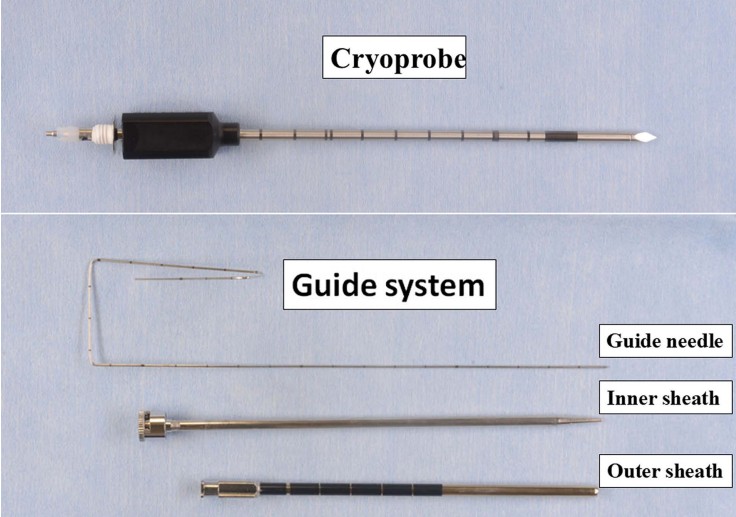

**Fig 1. A 3.4-mm diameter cryoprobe, and a guide needle system consisting of a guide needle and inner and outer sheaths.**

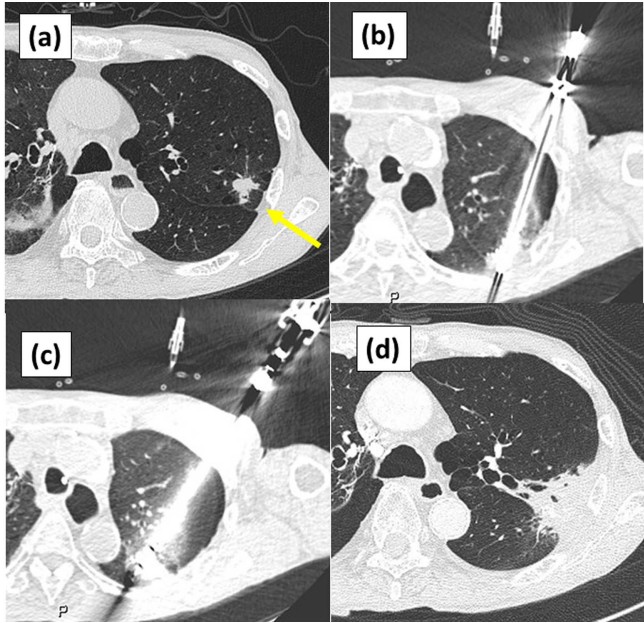

**Fig 2. Cryoablation procedure. (a)** A tumor is visible in the dorsal segment of the left upper lobe (arrow). First, **(b)** guide needle system is used for the outer sheath to penetrate the tumor, then **(c)** the cryoprobe is introduced into the outer sheath, followed by cryoablation. **(d)** Follow-up computed tomography images 5 months after treatment.

### Preserved pulmonary function evaluations

$FEV_1$ and forced vital capacity were measured before and 6 months after treatment. The percentage of preserved pulmonary function was calculated as follows: post-$FEV_1$/pre-$FEV_1 \times 100$ (%).

### Variables

Age, sex, comorbidities, a maximum standardized uptake value (SUV) on PET, BED, and the tumor size, stage, and histology were examined for correlations with LC and OS. Comorbidities were assessed using the Charlson Comorbidity Index [19]. Tumor size was defined as the maximum size including ground glass opacity area.

### Local recurrence

Local recurrence was defined as recurrence within the primary site with or without a pathological diagnosis. Other recurrences, including metastases to the hilar/mediastinal lymph nodes or lungs outside the primary site, were defined as distant metastases.

### Study outcomes

The primary outcomes were LC, OS, recurrence-free survival (RFS), disease-specific survival (DSS), and adverse events. The secondary outcome was a change in pulmonary function.

### Statistical analyses

The Kaplan–Meier method was used to evaluate the LC and survival outcomes. Univariate Cox proportional hazards regression analyses were used to identify correlations of variables with LC and OS. Variables with P-values of <0.05 in

the univariate analysis were entered into the multivariate analysis. The optimal cut-off value for significant variables on the multivariate analysis was determined using a receiver-operating characteristic (ROC) curve and Youden's index. The Wilcoxon signed-rank test was used to compare $FEV_1$ values before and after treatment. Statistical significance was set at P-values of <0.05. Statistical analyses were performed using SPSS Statistics version 29 (IBM Corp., Armonk, NY, USA).

## Results

Fig 3 presents the patient inclusion flow diagram. From 2015 to 2023, 220 patients with stage I NSCLC tumors ≥2 cm underwent local treatments, of whom 122, 20, and 14 underwent only surgery, SBRT, or cryoablation, respectively, and were excluded. In total, 64 patients underwent combined SBRT/cryoablation treatment (Table 1). Tumor stage was T1mi or T1a in 3 patients, T1b in 17, T1c in 32, and T2a in 12. Sixty-one patients (95%) underwent an FDG-PET. The combined SBRT/cryoablation treatment was administered to 28 patients (44%) due to a Charlson Comorbidity Index of ≥3, or $FEV_1$/FVC of <70%, or age of ≥80 years, whereas 36 patients (56%) patients without those unfavorable factors requested the combined treatment instead of surgery.

Table 2 presents the treatment details. For SBRT, the median fraction was 5 (interquartile range [IQR], 5–5), median total dose was 50 Gy (IQR, 44–50), and median BED was 100 Gy (IQR, 82–100). For cryoablation, the median number of probes was 1 (IQR, 1–1), and the median number of cycles was 3 (IQR, 3–4).

No patient was lost to follow-up. The median follow-up period was 74 months (range, 3–111; IQR, 41–89 months). In total, 15 (23%) patients relapsed, including local recurrence, distant recurrence, and both (2, 10, and 3 patients, respectively); recurrence did not occur in 49 (77%) patients. One patient died 3 months after the treatment due to non-treatment-related liver cirrhosis.

Fig 4 presents the Kaplan–Meier curves for LC, OS, RFS, and DSS. The 3- and 5-year LC rates were both 93% (95% confidence interval [CI], 87–100%; Fig 4a). The median LC was 73 months (range, 3–110; IQR, 36–87 months). Local recurrence was diagnosed in 5 (8%) patients, with a median LC of 16 months (range, 8–61; IQR, 13–34 months) (S1 Table).

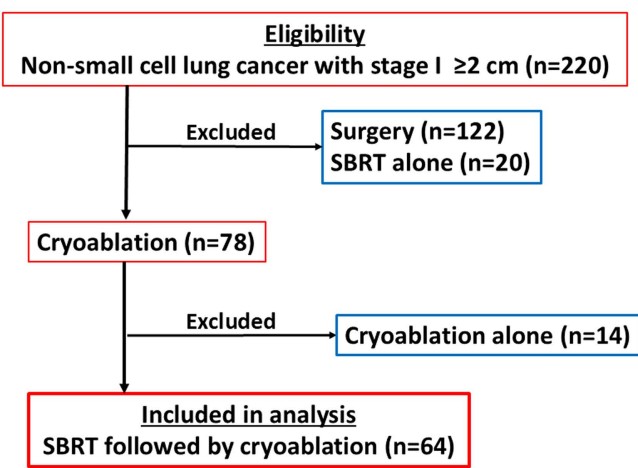

**Fig 3. Consort flow diagram.** In total, 78 of 220 patients with stage I non-small cell lung cancer tumors ≥2 cm underwent cryoablation, of which 64 were treated by SBRT followed by cryoablation. SBRT, stereotactic body radiation therapy.

**Table 1. Patient characteristics.**

| Characteristics | Value |
|---|---|
| Number of patients | 64 (100%) |
| Age (year) | 76 ± 7 (range: 58–89) |
| Sex | |
| Male | 41 (64) |
| Female | 23 (36) |
| $FEV_1/FVC$ (%) | 66 ± 13 (range: 33–89) |
| $\%FEV_1$ | 96 ± 30 (range: 38–184) |
| Charlson Comorbidity Index | 2.2 ± 1.3 (range: 0–7) |
| Tumor histology | |
| Adenocarcinoma | 41 (64) |
| Squamous cell carcinoma | 23 (36) |
| Tumor size (cm) | 2.7 ± 0.5 (range: 2.0–4.0) |
| 2–3 | 48 (75) |
| >3–4 | 16 (25) |
| T-stage | |
| T1mi or T1a | 3 (5) |
| T1b | 17 (27) |
| T1c | 32 (50) |
| T2a | 12 (18) |
| SUV on PET | 4.9 ± 4.2 (range: 0.6–17) |

$FEV_1/FVC$, forced expiratory volume in 1 second / forced vital capacity;

SUV, standardized uptake value; PET, positron emission tomography

**Table 2. SBRT and cryoablation treatment details.**

| SBRT | |
|---|---|
| Number of fractions | |
| Median (range) | 5 (4–10) |
| IQR | 5–5 |
| Total dose (Gy) | |
| Median (range) | 50 (40–60) |
| IQR | 42–50 |
| Biological effective dose (Gy) | |
| Median (range) | 100 (80–120) |
| IQR | 81–100 |
| V20 (%) | |
| Median (range) | 3.1 (1.5–4.5) |
| IQR | 2.3–4.0 |
| Cryoablation | |
| Number of cryoprobes used per procedure | |
| Median (range) | 1 (1–2) |
| IQR | 1–1 |
| Number of cycles per procedure | |
| Median (range) | 3 (3–5) |
| IQR | 3–4 |

SBRT, stereotactic body radiation therapy; IQR, interquartile range.

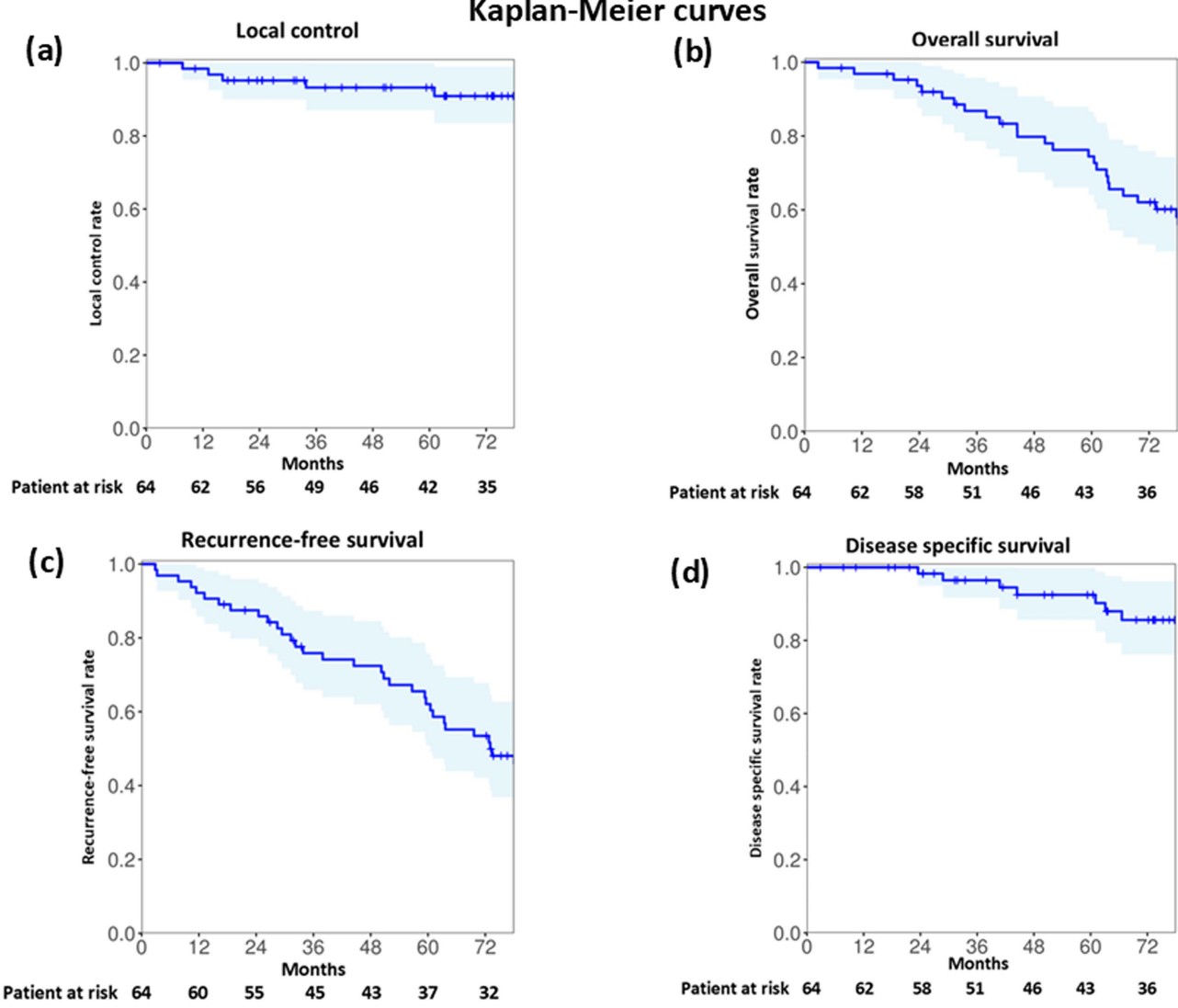

**Fig 4. Kaplan–Meier curves for 3- and 5-years for (a) local control (both 93%); (b) overall survival (87% and 74%, respectively); (c) recurrence-free survival.** (74% and 60%, respectively); and **(d)** disease-specific survival (96% and 92%, respectively). The shaded area indicates the 95% confidence interval.

The 3- and 5-year OS rates were 87% (CI, 78–95%) and 74% (CI, 63–86%), respectively (**Fig 4b**). The median OS was 74 months (range, 3–110; IQR, 41–89 months). During the follow-up period, 31 (48%) patients died; 7 due to primary disease (median OS: 44 months [range, 24–89; IQR, 35–65 months]) and 24 due to other diseases (median OS: 60 months [range, 3–96; IQR, 33–73 months]). Of the latter 24 patients, 18 died from non-cancer diseases, 5 from cancer of other organs, and 1 from secondary lung cancer.

The 3- and 5-year RFS rates were 74% (CI, 63–85%) and 60% (CI, 48–73%), respectively (**Fig 4c**). The median RFS was 64 months (range, 3–110; IQR, 32–82 months). The median RFS of the 15 patients who relapsed was 29 months (range, 3–96; IQR, 33–73 months), whereas that of the 49 patients who did not relapse was 75 months (range, 3–110; IQR, 44–89 months).

The 3- and 5-year DSS rates were 96% (CI, 92–100%) and 92% (CI, 85–100%), respectively (**Fig 4d**). The median DSS was 74 months (range, 3–110; IQR, 41–89 months). The median DSS of the 7 patients who died due to the primary disease was 53 months (range, 24–89; IQR, 38–64 months), whereas that of the 24 patients who died due to other diseases was 61 months (range, 3–110; IQR, 37–78 months).

Treatment-related mortality was not observed. Pneumothorax with CTCAE grade 2 occurred in 26 (40%) patients after cryoablation, with a median chest tube drainage period of 2 days (IQR, 1–4). Five (8%) patients developed radiation pneumonitis (CTCAE grade 2 in 4 patients and grade 3 in 1 patient) between 7 and 15 weeks after treatment, all of whom recovered without sequelae and survived 57–87 months after treatment (S2 Table). None of the patients experienced late-onset adverse events associated with SBRT, including interstitial pneumonitis, rib fracture, or pleural effusion.

The univariate Cox regression analysis did not identify any LC-related variables (P=0.22–0.63; S3 Table). However, significant correlations with OS were identified for the tumor stage, SUV on PET, and BED (all P=0.02). The tumor stage and BED remained significant variables in the multivariate analysis (P=0.04 and 0.02, respectively; Table 3). The ROC curve analysis identified optimal cut-off values of T1c and 81 Gy for the tumor stage and BED, respectively, with both area under the curve of < 0.7 (0.67 and 0.58, respectively).

Pulmonary function tests were performed before and 6 months after treatment in 59 of 64 patients (92%). $FEV_1$ significantly differed between pre- and post-treatment (1.8±0.6 vs. 1.7±0.5 L, P<0.001), with a mean post-/pre-$FEV_1$ percentage of 95±10% (median, 96; IQR, 90–100) (S1 Fig).

## Discussion

This study has four key findings. First, the 5-year OS rate for stage I NSCLC with tumors ≥2 cm after combined SBRT/cryoablation treatment was better than that of previous studies after SBRT alone for stage I NSCLC including tumors <2 cm. Second, the incidence of radiation pneumonitis was similar to that after SBRT alone. Third, the decrease in pulmonary function was comparable to that after SBRT alone. Finally, pneumothorax was observed after cryoablation in 40% of patients, with a median drainage period of 2 days.

Most studies on SBRT alone have reported 3-year LC rate of >90% [1–4], consistent with our findings of 93%. The RTOG 0618 trial reported a 4-year LC rate of 96% [5]. However, high LC rates after SBRT can be misleading owing to potential residual tumors hidden in the large scars that form after SBRT, which could cause distant metastases. Therefore,

**Table 3. Cox proportional hazard regression analyses for overall survival.**

| Variables | Univariate analysis | | Multivariate analysis | |
|---|---|---|---|---|
| | HR (95% CI) | P-value | HR (95% CI) | P-value |
| Age | 1.0 (1.0–1.1) | 0.25 | — | |
| Sex | 0.7 (0.3–1.5) | 0.38 | — | |
| Comorbidity index | 1.2 (0.9–1.5) | 0.17 | — | |
| Tumor size | 1.6 (0.8–3.0) | 0.17 | — | |
| Tumor stage | 1.7 (1.1–2.8) | 0.02 | 1.8 (1.0–3.2) | 0.04 |
| SUV on PET | 1.1 (1.0–1.2) | 0.02 | 1.0 (1.0–1.1) | 0.32 |
| Tumor histology (Ad vs. Sq) | 1.9 (0.9–3.8) | 0.09 | — | |
| BED | 1.0 (0.9–1.0) | 0.02 | 1.0 (0.9–1.0) | 0.02 |

HR, hazard ratio; CI, confidence interval;

SUV, standard uptake value; PET, positron emission tomography;

Ad, adenocarcinoma; Sq, squamous cell carcinoma;

BED, biological effective dose.

the high LC rates after SBRT, including that in the present study, may be unreliable. Treatment outcomes after SBRT should be evaluated based on OS, not on LC.

In this study, the 3-year OS rate after the combined SBRT/cryoablation treatment was 87%, which was equal to or higher than that previously reported rates for SBRT alone (43–83%) [1–4]. However, 3 years is too short to estimate the OS of patients with stage I NSCLC; OS follow-up should be ≥5 years. The RTOG 0618 and RTOG 0236 studies, as well as the study by Shibamoto et al., reported 5-year OS rates of 41%, 40%, and 52%, respectively, after SBRT alone for stage I NSCLC including tumors <2 cm [5–7]. In contrast, the 5-year OS rate was 75% in our study for tumors ≥2 cm, suggesting that the combined SBRT/cryoablation treatment can cure primary tumors and prevent distant recurrences more effectively than SBRT alone. The median OS in the 24 patients who died of other diseases was 60 months, which did not considerably affect the 74-month median OS of the entire cohort.

BED was significantly associated with OS in the multivariate analysis, with a cut-off value of 81 Gy. The patients treated with BED lower than 100 Gy were older than 80 years or had interstitial pneumonitis on CT. Although the area under the ROC curve for defining the cut-off value was just 0.58, indicating a low accuracy, the desirable BED would be close to 100 Gy.

Although this study included operable and inoperable patients, age and comorbidity were not associated with OS, emphasizing the generalizability of the combined SBRT/cryoablation treatment for both patient groups.

Surgery is the standard treatment for stage I NSCLC. A nationwide lung cancer registry study of > 11,000 surgical cases reported that the 5-year OS rates for stages IA and IB were 82% and 67%, respectively [20], which is comparable to the 75% 5-year OS rate in the present study for stage I NSCLC with tumors ≥2 cm.

Before this study, a primary concern was that cryoablation might increase the incidence of radiation pneumonitis. Radiation pneumonitis of grade 2 or higher occurred in 5 patients (8%) in this study, consistent with the reported incidence following SBRT alone (7–13%) [3,4,7,21]. This result suggests that cryoablation does not cause further inflammation after SBRT.

To clarify the severity of lung damage, pulmonary function was assessed before and after treatment. We found that the pulmonary function at 6 months after treatment was 95% of the pretreatment value. This result is comparable to those reported after SBRT alone (92–96%) [22–24]. Therefore, the combined SBRT/cryoablation treatment did not induce greater lung damage compared to SBRT alone.

Pneumothorax, a known complication of cryoablation and RFA, occurred in 40% of patients in the present study. However, the median drainage period was only 2 days; it was not considered a severe adverse event.

This study has some limitations. First, the study design was retrospective, included only 64 patients, and lacked a control group, which impeded the establishment of causal inference between the treatment and outcome. Second, a biopsy of the primary tumor was not routinely conducted at the time of distant metastases; therefore, LC could not be precisely estimated.

## Conclusion

Combination of SBRT/cryoablation is a feasible treatment for stage I NSCLC with tumors ≥2 cm. However, as this study was a single-arm cohort study, it cannot be directly compared with studies of SBRT alone, all of which were also single-arm studies. Nonetheless, this study presents a novel treatment option for patients with stage I NSCLC tumors ≥2 cm.

## Supporting information

**S1 Fig. Box plot comparison of $FEV_1$ pre- and 6-months post-treatment.** $FEV_1$ significantly differs between the pre- and post-treatment (1.8±0.6 vs. 1.7±0.5 L, P<0.001), with a mean post-/pre-$FEV_1$ percentage of 95±10% (median: 96; interquartile range: 90–100). $FEV_1$, forced expiratory volume in 1 second.
(TIF)

**S1 Table. Characteristics of patients with local recurrence.**
(DOCX)

**S2 Table. Characteristics of patients with pneumonitis.**
(DOCX)

**S3 Table. Cox proportional hazard regression analysis for local recurrence.**
(DOCX)

## Author contributions

**Data curation:** Hiroaki Nomori, Yue Cong.

**Investigation:** Hiroaki Nomori, Yue Cong, Hiroyoshi Iguchi, Kenichi Kashihara, Ryuichi Wada, Tsutomu Saito.

**Methodology:** Hiroaki Nomori, Hiroyoshi Iguchi, Kenichi Kashihara, Tsutomu Saito.

**Writing – original draft:** Hiroaki Nomori.

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
