## [Decision Letter · Decision Letter 0]

14 May 2025

PONE-D-25-05871Long-term outcomes of combination therapy with stereotactic body radiation therapy and cryoablation for T1-2N0M0 non-small cell lung cancer with tumor size ≥2 cmPLOS ONE

Dear Dr. Nomori,

Thank you for submitting your manuscript to PLOS ONE. After careful consideration, we feel that it has merit but does not fully meet PLOS ONE’s publication criteria as it currently stands. Therefore, we invite you to submit a revised version of the manuscript that addresses the points raised during the review process.

We look forward to receiving your revised manuscript.

Kind regards,

Xing-Xiong An, M.D.

Academic Editor

PLOS ONE

Journal Requirements:

Reviewers' comments:

Reviewer's Responses to Questions

**Comments to the Author**

1. Is the manuscript technically sound, and do the data support the conclusions?

Reviewer #1: Yes

Reviewer #2: Partly

Reviewer #3: Yes

Reviewer #4: Partly

Reviewer #5: Partly

2. Has the statistical analysis been performed appropriately and rigorously? 

Reviewer #1: Yes

Reviewer #2: I Don't Know

Reviewer #3: Yes

Reviewer #4: Yes

Reviewer #5: Yes

3. Have the authors made all data underlying the findings in their manuscript fully available?

Reviewer #1: No

Reviewer #2: No

Reviewer #3: No

Reviewer #4: Yes

Reviewer #5: Yes

4. Is the manuscript presented in an intelligible fashion and written in standard English?

Reviewer #1: Yes

Reviewer #2: Yes

Reviewer #3: Yes

Reviewer #4: Yes

Reviewer #5: No

5. Review Comments to the Author

Reviewer #1: The manuscript is claerly written, with figures supporting the findings. Please, clarify more precisely how internal margin was defined (lines 158-160) and explain the rationale for accepting a BED <100Gy.

Since bias related to observational and retrospective nature of the study is highlighted, it is bold to include a comparation with results of a prospective study on SBRT in the conclusion.

Moreover, I suggest emphazing the small size of cohort in the discussion.

Reviewer #2: Thank you for the opportunity to review this manuscript on the long-term outcomes of combined stereotactic body radiotherapy and cryoablation for NSCLC. This study is a retrospective descriptive study reporting long-term outcomes after combined stereotactic body radiotherapy and cryoablation for T1-T2N0 NSCLC. There are very few reports on a combined approach for the treatment of lung cancers or oligometastasis. While the approach is novel, several issues in this study need addressing to ensure scientific validity.

Major issues

1. Table 1: There is one patient with a T3 tumour (>5cm). If the inclusion is T1-T2, I am unsure why this patient was included. I suggest excluding T3 tumours.

2. A BED of at least 80 Gy will be considered low according to the current recommendation of at least 100 Gy, as outcomes tend to be poorer at doses less than 100 Gy. It would be beneficial to discuss the reasons why some tumours were treated with doses below 100 Gy. It is unclear whether the dose was reduced to facilitate combined treatment or to meet OAR dose constraints.

3. The follow-up CT scans ceased at three years (page 11, line 181), despite a median follow-up of 73 months. No information has been provided regarding the methods used to diagnose recurrences after this period.

4. Local recurrences were diagnosed either through needle biopsy or when “the primary site showed apparent increase” (page 12, line 208-9). Different patterns of fibrosis, including dense mass-like consolidation, can be observed after SBRT, making the criteria used in the study for radiologically diagnosing recurrence inadequate. It is necessary to provide a clear description of the criteria used for radiological diagnosis of recurrence when a needle biopsy was not performed, including the criteria used to select patients for needle biopsy and how radiation pneumonitis was differentiated from recurrence.

5. In the regression analysis of factors correlating with LC and survival, BED of SBRT should be included, since it has been shown to affect these outcomes.

6. It has been noted that the majority of SBRT studies do not report results beyond three years, which is accurate. However, RTOG 0915 and RTOG 0236 have presented 5-year outcomes, with LC rate of 89.4-93.2%, and RTOG 0618 presented a 4-year LC of 96%. The local control rates observed in these RTOG studies are comparable to those reported in this study. I suggest discussing the results within the context of the outcomes of the RTOG studies.

7. 24 out of 32 patients died due to "other disease" (page 15, line 248-9). The high rate of competing non-cancer mortality can diminish the incidence of cancer-related events, which has been an issue in comparing SBRT studies. It is essential to discuss how competing non-cancer mortality affects the assessment of treatment outcomes.

8. In the conclusion, it is inappropriate to suggest that combined treatment improved survival, as this is a single-arm cohort study without a comparison group. Furthermore, stating that the rates of adverse events were acceptable significantly understates the 38% incidence of pneumothorax requiring chest drainage and possibly hospitalisation. This incidence does not compare favorably with the low occurrence of adverse events observed following SBRT alone. Conclusions should take into account the significant limitations of the study.

9. Comparing Kaplan-Meier graphs for OS and DSS (Fig. 4 and 6), the number of patients at risk is nearly identical in both graphs. Given that 75% of the deaths were due to non-cancer causes, there should be a notable difference in the numbers at risk. The statistical analyses were performed using Excel, which I understand does not have a built-in Kaplan-Meier graph function and thus requires a third-party add-in or some manual steps. I recommend verifying the results using alternative statistical software.

Minor issues

1. T1-2 tumour with tumour size > 2cm as per 8th edition will be T1c-T2. I suggest using T1c-T2 instead of a lengthier description.

2. Abstract: The last line in the Methods (page 3, line 50), “Median follow-up duration was 73 months”, is a repetition since the median follow-up is mentioned in the first paragraph (line 42).

3. Table 1: The breakdown of T stage % adds up to only 91%.

4. Tables 3 and 4: the last column representing the cause of mortality is labelled “Prognosis, " which refers to the chance of an event happening. I suggest labelling it mortality and mentioning whether it is due to cancer or other causes for each patient.

5. The terms pneumonia and pneumonitis are used interchangeably to describe radiation pneumonitis. I suggest using radiation pneumonitis for consistency. This applies to tables 4 and 7 as well.

6. I suggest labelling Fig. 2 as a Consort flow diagram instead of “ patient algorithm”. Use terms such as assessed for eligibility, excluded, included in analysis, etc, to label each box clearly.

7. I suggest labelling the last subheading as Discussion rather than Comments.

8. There is use of the term “other disease” in several places, which I presume refers to a disease other than cancer. However, I am unsure if it excludes second cancers. This needs clarification.

Reviewer #3: Line 100-102: Sentence not clear.

Line 131-136: I don’t see any information on setting.

Line 135: Are those numbers in the brackets dates or what?

Line 146: What cut off was classified as old age?

Line 148-149: Did you have to test before determining if tumor was outside the reach of the probe?

Be consistent with the use of your terms. Mixing terms like local recurrence, survival, overall survival, local control makes it confusing.

Line 222: P≤0.15?? Why?

Line 233-236: Were there multiple reasons for participation? Numbers seem to exceed 66 by a lot.

Line 239 and 241: IQR should be checked.

Line 243: You followed some participants for 3 months. What happened if they were not lost to follow-up? Were they deceased?

Line 268: Is it the traditional p-value? Seems unusually large. Any justification for that?

Line 269: You set p value at 0.05, so why do you consider 0.09 significant

Line 282: What do you mean by comment?

You could explore the influence of patient’s prognosis on the outcomes. Poor prognosis can most likely after survival.

You need to provide a more thorough discussion for your findings.

Line 320-324: Don’t just state the limitations of the studies but proceed to provide suggestions for future steps.

Work on conclusion and make it better.

Reviewer #4: This study investigates a relatively uncommon treatment strategy—SBRT followed by cryoablation—for early-stage NSCLC patients who are ineligible for surgery. Reporting favorable 5-year outcomes in this population is noteworthy and clinically interesting, especially given the limited data available for this specific combination approach. The study’s strength lies in its long follow-up period and the effort to explore a minimally invasive option for patients who are not surgical candidates.

However, despite the clinical relevance and the encouraging outcomes, the lack of a control group, unclear patient selection criteria, and potential selection bias make it difficult to draw strong conclusions about the superiority or added benefit of the combined approach. The conclusions, as currently written, overstate the findings beyond what the data can support. Meaningful revisions to the manuscript are necessary to improve clarity, transparency, and scientific robustness.

- Justification for SBRT and Cryoablation Strategy

The rationale behind applying cryoablation after SBRT is not adequately justified. Authors suggest that OS might improve by preventing distant recurrence or eradicating residual tumor, but this hypothesis requires more mechanistic and clinical justification.

It is recommended that the Introduction and Discussion sections provide a more detailed and logically structured explanation of why this combination strategy was pursued. Ideally, this justification should be supported by relevant preclinical or clinical literature. Including references that explain potential synergistic effects or biological rationale would strengthen the scientific foundation of the study design.

- Patient Selection and Treatment Allocation

It is unclear how the treatment modality was determined among 333 patients. What were the specific indications for: Surgery, SBRT alone, Cryoablation alone, or SBRT followed by cryoablation. This ambiguity severely limits the interpretability of the study results and introduces significant selection bias.

For example, cryoablation could only be performed for “tumors located within the reach of the cryoprobe,” suggesting technical feasibility was a selection factor. This should be explicitly acknowledged and addressed as a source of bias.

- Need for Comparator Group

The study would benefit greatly from a matched SBRT-alone cohort to determine whether the additional cryoablation offers real benefit in terms of OS, LC, or adverse events. Without such a comparison, the claim that combination therapy is superior to SBRT alone is not supportable.

A statement like “SBRT followed by cryoablation improves survival compared to SBRT alone” (e.g., lines 60–63, 328–330) is speculative and not supported by direct comparison in this study. The study lacks a control group of patients receiving SBRT alone. Thus, the conclusion should be softened (e.g., “SBRT followed by cryoablation showed favorable outcomes…”).

- Patterns of Recurrence

It would significantly strengthen the paper to report patterns of recurrence (local, regional, or distant) and provide detail on subsequent treatments received, especially in recurrent cases.

- Toxicity Assessment

The distinction between bronchopneumonia (BP) and radiation pneumonitis (RP) is not clearly explained. What were the diagnostic criteria used to differentiate the two?

CTCAE version 4.03 is stated to be used in the main text, but Reference 17 refers to CTCAE v3.0, which should be corrected.

- Overuse of Figures and Tables

Consider combining Figures 3–6 into a single multipanel survival figure (e.g., Kaplan-Meier curves).

Tables 3–5 may be more appropriate as Supplementary Tables, as they do not add substantial value to the main narrative.

- Outdated References

Many references are more than a decade old. Recent data on SBRT, cryoablation, and combination therapies should be incorporated to better contextualize the study findings.

Reviewer #5: The manuscript presents a well-designed retrospective cohort study on the long-term outcomes of combined SBRT and cryoablation for cT1-2N0M0 NSCLC with tumor size ≥2 cm. The median follow-up of 73 months and focus on a clinically relevant population – patients ineligible or unwilling to undergo surgery – enhance the significance of this work. The results are clearly reported and show promising local control and overall survival outcomes.

However, several areas require clarification, elaboration, and improvement to strengthen the manuscript's scientific rigor and clinical impact. Please refer to the attached documents of suggested revision.

6. PLOS authors have the option to publish the peer review history of their article (what does this mean? ). If published, this will include your full peer review and any attached files.

**Do you want your identity to be public for this peer review?** For information about this choice, including consent withdrawal, please see our Privacy Policy .

Reviewer #1: No

Reviewer #2: No

Reviewer #3: No

Reviewer #4: No

Reviewer #5: No

---

## [Author Response · Author response to Decision Letter 1]

10 Jun 2025

June 6, 2025

[Editor’s name]　Dr. Xing-Xiong An, M.D

[Journal name」PLOS ONE

Dear Editor:

We wish to re-submit our revised manuscript titled, “Long-term outcomes of stereotactic body radiation therapy followed by cryoablation using liquid nitrogen for stage I non-small cell lung cancer with tumor size ≥2 cm.” The manuscript ID is PONE-D-25-05871.

We thank you and the reviewers for your thoughtful suggestions and insights. The manuscript has benefited from these helpful comments, and we look forward to working with you and the reviewers to move this manuscript closer to publication.

The manuscript has been rechecked and the necessary changes have been made in accordance with the reviewers’ suggestions. We hope these meet the approval of the reviewers. These corrections do not influence the content and framework of the paper. The responses to all comments have been prepared and attached given below.

We have marked them in Red Bold type in the marked version of manuscript.

We have also attached the non-marked one.

Thank you for your consideration. We look forward to hearing from you.

Sincerely,

Hiroaki Nomori, MD

Department of General Thoracic Surgery, Kashiwa Kousei General Hospital, 617 Shikoda, Kashiwa city, 277-8551, Chiba, Japan

Tel: +81-471-45-1111

E-mail: hnomori@qk9.so-net.ne.jp

Response to reviewer

The corrected points were in marked paper with Red Bold type.

1. Novelty and Contextualization

The rationale for combining SBRT and cryoablation is interesting but not fully contextualized in relation to prior combination therapy reports (if any exist). A more explicit explanation of how this study is novel compared to previous SBRT or ablation-alone studies is warranted.

➡　From 37 – 49: We rewrote the Objectives in the abstract as follows: While radiation and ablation therapy have been performed for stage I non-small cell lung cancer, their local control and survival rates are not satisfactory for tumors ≥2 cm. This study administered a combination therapy of stereotactic body radiation and cryoablation and evaluated its effectiveness based on 5-year survival rates in patients with stage I non-small cell lung cancer ≥2 cm.

While the radiation or ablation therapy has been conducted for T1N0M0 lung cancer, most of their survival have been examined with 3-year, which is not long enough. In fact, the 5-year survival of stereotactic body radiation for stage I lung cancer was reportedly decreased to 40%. This study aimed to assess 5-year treatment results with combination therapy of stereotactic body radiation and cryoablation for T1-2N0M0 non-small cell lung cancer with tumor size ≥2 cm with 73 months of median follow-up.

➡From line 121 – 131: We wrote the following:

However, the LC rates after both cryoablation and RFA are reportedly lower for NSCLCs with tumor size ≥2 cm than those <2 cm [10,11], consistent with SBRT results [9].

The mechanism of tumor control differs for SBRT, cryoablation, and RFA. SBRT causes injury to the tumor DNA. Cryoablation induces the formation of ice crystals, leading to cell membrane destruction and subsequent intracellular dehydration. RFA leads to coagulation necrosis owing to high temperatures. Due to the different mechanisms, a combination of these therapies may facilitate improve LC for stage I NSCLC with tumor size ≥2 cm; however, this combination therapy has not been explored.

2. Patient Selection

• The study mixes patients who were medically inoperable and those who preferred non-surgical treatment. This creates heterogeneity in the cohort. Please discuss how this choice affects generalizability and whether outcomes differ between these two groups.

➡From line 368 – 370: We wrote the following:

Although the study included both operable and inoperable patients, the OS exhibited no correlations with age and comorbidity, indicating the generalizability of the SBRT-cryoablation treatment for both patient groups.

• Inclusion of two patients with stage II NSCLC (T2bN0M0) should be acknowledged and justified more explicitly or excluded for consistency with the stated focus on Stage I disease.

➡ We deleted 2 patients with T2b, i.e., stage II, which changed the manuscript as follows:

From line 1 – 3: Title was changed to the following:

“Long-term outcomes of stereotactic body radiation therapy followed by cryoablation using liquid nitrogen for stage I non-small cell lung cancer with tumor size ≥2 cm”.

From line 37 – 41: The abstract was changed to the following:

While radiation and ablation therapy have been performed for stage I non-small cell lung cancer, their local control and survival rates are not satisfactory for tumors ≥2 cm. This study administered a combination therapy of stereotactic body radiation and cryoablation and evaluated its effectiveness based on 5-year survival rates in patients with stage I non-small cell lung cancer ≥2 cm.

While the radiation or ablation therapy has been conducted for T1N0M0 lung cancer, most of their survival have been examined with 3-year, which is not long enough. In fact, the 5-year survival of stereotactic body radiation for stage I lung cancer was reportedly decreased to 40%. This study aimed to assess 5-year treatment results with combination therapy of stereotactic body radiation and cryoablation for T1-2N0M0 non-small cell lung cancer with tumor size ≥2 cm with 73 months of median follow-up.

From line 50-52: The number of patients was changed as “64 patients with stage I non-small cell lung cancer ≥2 cm 66 patients with T1-2N0M0 non-small cell lung cancer ≥2 cm”

Line 58: The median follow-up duration was changed from 73 to 74 months.

Line 59; The mean tumor diameter was change to 2.7 ± 0.5 0.7 (range: 2.0–4.0 5.4) cm.

Line 61: The 5-year overall survival was changed from 75 to 74%.

Line 63: post-cryoablation pneumothorax was changed from 38 to 40%.

Line 64: The number of patients with pneumonitis was changed from 6 (9%) to “Five (8%)”

Line 65: The number of patients with radiation pneumonitis was changed from 5 to 4.

Line 66: The number of patients with grade 2 pneumonitis was changed from five to four.

Line 69: T1-2N0M0 was change to stage I.

Line 145 - 146: The number of patients was changed to 64 patients with stage I 66 patients with cT1-2N0M0 .

Line 148: Median follow-up period was changed from 73 to 74 months.

Line from 278 – 286: The patient algorithm was changed as the following:

Between 2015 and 2023, 220 patients with stage I NSCLC with tumor size ≥2 cm underwent local treatments, including surgery, SBRT, and cryoablation (Fig. 3). Among the 78 patients who underwent cryoablation, 64 patients were treated with the SBRT-cryoablation regimen (Table 1). 333 patients with cT1-2N0M0 non-small cell lung cancer with tumor size ≥2 cm underwent local treatments, including surgery, SBRT, and cryoablation (Fig. 2). Among 80 patients who underwent cryoablation, 66 patients, including 2 patients with T2bN0M0 (Stage II), were treated by SBRT followed by cryoablation (Table 1).

Line from 291 – 292: It was changed as the following: “The FDG-PET was conducted for 61 of the 64 patients (95%) 62 of the 66 patients (94%).”

Line from 298 – 299: It was changed as the following: The median follow-up period was 74 (range, 3–111; IQR, 41–89) months. 73 months (range, 3–111; IQR, 42–88).

Line from 303 – 305: It was changed as the following: “The 3- and 5-year OS rates were 87% (CI, 78–95% 79%–96%) and 74% (CI, 63–86% 64%–86%), respectively (Fig. 5 4). During the follow-up period, 31 (48%) 32 (48%)died, of which 23 24 died due to other diseases.

Line from 306 – 309: The RFS was changed as the following: “The 3- and 5-year RFS rates were 74% (CI, 63–85% 63%–84%) and 60% (CI, 48–73%), respectively (Fig. 6 5). Twelve patients (14%) Fifteen patients (23%) experienced distant metastases between 3 and 60 months after treatment.

Line from 310 – 311: The DSS was changed as “The 3- and 5-year DSS rates were 96% (CI, 92–100%) and 92% (CI, 85–100%) 97% (CI, 92%–100%) and 93% (CI, 86%–100%), respectively (Fig. 7 6).“

Line from 313 – 319: It was changed as the following: Pneumothorax with CTCAE grade 2 occurred in 26 (40%) in 25 (38%) patients patients after cryoablation, with a median chest tube drainage period of 2 days (IQR, 1–4). Median duration of hospital stay after cryoablation was 5 days (IQR, 4–5). Five (8%) Six (9%) patients developed pneumonia with CTCAE grade 2 (n=4) and grade 3 (n=1) between 7 and 15 weeks after treatment (Table 4). Among them, four five had radiation pneumonitis and one had bronchopneumonia. ”

Line from 334 – 339: It was changed as the following: “Pulmonary function tests were performed before and 6 months after treatment in 59 of 64 patients (92%) 61 (92%) 66 patients. A significant decrease in FEV1 was observed between pretreatment and posttreatment (1.8 ± 0.6 vs. 1.7 ±0.5 0.6 L, P < 0.001), of which the mean post-/pre-FEV1 percentage was 95 ± 10% (median, 96; IQR, 90–100) (Fig. 8). 96 ± 9% (median, 97; IQR, 92–102) (Fig. 7).

From line 342 – 346: It was changed as “This study highlights the following points: (1) the 5-year OS after SBRT-cryoablation regimen for stage I NSCLC ≥2 cm was better than that reported in previous studies on SBRT alone for stage I NSCLC including tumors < 2 cm 5-year OS after SBRT followed by cryoablation was better than that reported in studies on SBRT alone; ”

Line 348-350: It was changed as “ (4) pneumothorax after cryoablation was observed in 40% 38% of the patients, of which median drainage period was 2 days.

3. SBRT Technique Clarification

➡From 175 – 207: SBRT technique was written in detail as follows: “SBRT was administered using an Elekta Versa HD (Elekta Co. Ltd., Tokyo, Japan). Respiratory motion was managed using three sets of conventional free-breathing CT simulation images, with an abdominal compression device for breath-hold.

Internal tumor volume (ITV), encompassing the full range of tumor motion, was delineated using the second and the third image sets and referenced to the clinical target volume (CTV) defined in the first image set. A planning target volume (PTV) margin of 5 mm was added to the ITV based on the Radiation Therapy Oncology Group (RTOG) studies 0236, 0813, and 0915.

Fractionation schemes were determined by radiation oncologists to optimize the PTV coverage while adhering to organ-at-risk dose constraints based on the RTOG protocols. Each fractionation schedule delivered a biological effective dose (BED) of at least 80 Gy (α/β=10 Gy).

Treatments were administered once daily on consecutive days. Before each irradiation session, a flat-panel cone-beam CT scan was performed to identify the tumor location. This was compared to the CTV from the treatment plan. If a positional discrepancy was detected, corrections were made using a six-degree-of-freedom treatment couch or manually.

Respiratory motion was managed using three sets of conventional free-breathing CT simulation images, with or without the use of a breath limitation device.

Internal tumor volume encompassing the full range of tumor motion was delineated using the second and the third images of the clinical target volume of the first image. A planning target volume expansion of 5 mm was applied to the internal tumor volume. Fractionation schemas were determined by radiation oncologists to optimize the planning target volume coverage and meet organ-risk dose constraints based on Radiation Therapy Oncology Group (RTOG) 0236, 0813, and 0915 trials. Each fractionation schedule contained a biological effective dose (BED) of at least 80 Gy (α/β=10). Treatment courses were administered using a consecutive-day, one-daily fractionation schedule.

• The SBRT section lacks clarity on how tumor motion was handled quantitatively. Was 4D-CT used? Are the three free-breathing scans equivalent to a poor-man's ITV?

➡All CT scans used in this study were volumetric (3D) scans, which capture the tumor three-dimensionally. While we did not use 4D-CT, we believe that acquiring and fusing three separate CT scans taken at different times can serve as a practical substitute for assessing tumor motion over time. To minimize respiratory variability and avoid a so-called “poor-man’s ITV,” patients were instructed to maintain consistent breathing during all scans. This instruction is a standard procedure at all treatment facilities involved and was therefore not specifically mentioned in the manuscript.

• The use of a 5 mm margin to define PTV may be standard, but justification or comparison with motion management strategies (e.g., gating, abdominal compression) would be helpful.

➡We used abdominal compression with a belt to reduce respiratory motion.

4. Statistical Analysis

• Multivariate analyses were conducted despite small event numbers (e.g., only 6 pneumonia events). Please clarify how overfitting or type II errors were mitigated, or consider simplifying the analysis or using Firth’s correction where applicable.

➡　From 330 – 333: We deleted the analysis for pneumonia and Table 7 as follows: Univariate logistic regression analysis for pneumonia showed correlations (p≤0.15) with age (p=0.15), tumor size (p=0.11), and V20 (p=0.14); however, these correlations were not significant in the multivariate analysis (p=0.18–0.51) (Table 7).

• The use of Microsoft Excel 10 for statistical analysis may not inspire confidence in reproducibility. Consider using a more robust statistical platform (e.g., R, SPSS, or Stata) or at least justify the use of Excel.

➡ From line 273 – 275: We used SPSS for statistical analysis as “Statistical analyses were performed using SPSS Statistics version 29 (IBM, NY, USA) using Microsoft Excel version 10 (Redmond, WA, USA).”

5. Reported events

• There is no grading of pneumothorax severity. Consider including this to align with CTCAE standards.

➡From line 313 – 315: We added the pneumothorax severity as “Pneumothorax with CTCAE grade 2 occurred in 26 (40%) in 25 (38%) patients patients after cryoablation, with a median chest tube drainage period of 2 days (IQR, 1–4).”

• The decrease in FEV1 is statistically significant but arguably not clinically meaningful. This distinction should be emphasized.

➡From line 382 – 383: We wrote the reason for examining decrease in FEV1 as “To clarify severity of lung damage, the pulmonary function was assessed both before and after treatment.”

While the manuscript aims to compare cryoablation and SBRT for early-stage NSCLC, it currently lacks sufficient detail regarding the cryoablation technique. For the study to be scientifically rigorous and clinically informative, it is important that the authors provide a comprehensive description of the cryoablation procedure. This should include, but not be limited to probe placement strategy, imaging modality for guidance, lesion size eligibility, and criteria for defining adequate ablation margins. Without this information, readers are unable to fully assess the comparability of the two treatment modalities, and the reproducibility of the cryoablation approach is compromised. I recommend that the authors include this information in the revised manuscript.

➡　Please see our reply for this in the next part.

6. Lack of details on the Cryoablation Technique

There is no information on imaging guidance used, lesion size limitation, or margin assessment. Without understanding how cryoablation was applied, it’s difficult for readers to assess if the outcomes were influenced by technical variability, operator experience, or treatment adequacy.

➡From line 217 – 225: We wrote the detailed technique of cryoablation as “The patient was placed in the supine or prone position on the CT table, depending on the tumor location. Sedatives such as pethidine hydrochloride (35 mg) and midazolam (2–3 mg) were administered. Following local anesthesia, the guide needle was introduced into a tumor under CT guidance (Fig. 2). The inner and outer sheaths were advanced over the guide needle to reach the tumor. Once the outer sheath reached the target position, the guide needle and inner sheath were removed. The cryoprobe was then introduced into the outer sheath, and freezing commenced.”

➡　We also added the CT views for placement the cryoablation and the CT 5 months after the trea

---

## [Decision Letter · Decision Letter 1]

3 Jul 2025

PONE-D-25-05871R1Long-term outcomes of stereotactic body radiation therapy followed by cryoablation using liquid nitrogen for stage I non-small cell lung cancer with tumor size ≥2 cmPLOS ONE

Dear Dr. Nomori,

Thank you for submitting your manuscript to PLOS ONE. After careful consideration, we feel that it has merit but does not fully meet PLOS ONE’s publication criteria as it currently stands. Therefore, we invite you to submit a revised version of the manuscript that addresses the points raised during the review process.

We look forward to receiving your revised manuscript.

Kind regards,

Xing-Xiong An, M.D.

Academic Editor

PLOS ONE

**Additional Editor Comments:**

Please provide point-by-point responses to each reviewer's comments, instead of just roughly putting all the answers together. 

For example:

**Response to reviewer #1:**

Comment 1:.....

Response 1:.....

Comment 2:.....

Response 2:.....

Comment 3:.....

Response 3:.....

……

**Response to reviewer #2:**

Comment 1:.....

Response 1:.....

Comment 2:.....

Response 2:.....

Comment 3:.....

Response 3:.....

……

**Response to reviewer #3:**

Comment 1:.....

Response 1:.....

Comment 2:.....

Response 2:.....

Comment 3:.....

Response 3:.....

……

**Response to reviewer #4:**

….

**Response to reviewer #5:**

….

**Previous Comments from Reviewers:**

Reviewer #1: The manuscript is claerly written, with figures supporting the findings. Please, clarify more precisely how internal margin was defined (lines 158-160) and explain the rationale for accepting a BED <100Gy.

Since bias related to observational and retrospective nature of the study is highlighted, it is bold to include a comparation with results of a prospective study on SBRT in the conclusion.

Moreover, I suggest emphazing the small size of cohort in the discussion.

Reviewer #2: Thank you for the opportunity to review this manuscript on the long-term outcomes of combined stereotactic body radiotherapy and cryoablation for NSCLC. This study is a retrospective descriptive study reporting long-term outcomes after combined stereotactic body radiotherapy and cryoablation for T1-T2N0 NSCLC. There are very few reports on a combined approach for the treatment of lung cancers or oligometastasis. While the approach is novel, several issues in this study need addressing to ensure scientific validity.

Major issues

1. Table 1: There is one patient with a T3 tumour (>5cm). If the inclusion is T1-T2, I am unsure why this patient was included. I suggest excluding T3 tumours.

2. A BED of at least 80 Gy will be considered low according to the current recommendation of at least 100 Gy, as outcomes tend to be poorer at doses less than 100 Gy. It would be beneficial to discuss the reasons why some tumours were treated with doses below 100 Gy. It is unclear whether the dose was reduced to facilitate combined treatment or to meet OAR dose constraints.

3. The follow-up CT scans ceased at three years (page 11, line 181), despite a median follow-up of 73 months. No information has been provided regarding the methods used to diagnose recurrences after this period.

4. Local recurrences were diagnosed either through needle biopsy or when “the primary site showed apparent increase” (page 12, line 208-9). Different patterns of fibrosis, including dense mass-like consolidation, can be observed after SBRT, making the criteria used in the study for radiologically diagnosing recurrence inadequate. It is necessary to provide a clear description of the criteria used for radiological diagnosis of recurrence when a needle biopsy was not performed, including the criteria used to select patients for needle biopsy and how radiation pneumonitis was differentiated from recurrence.

5. In the regression analysis of factors correlating with LC and survival, BED of SBRT should be included, since it has been shown to affect these outcomes.

6. It has been noted that the majority of SBRT studies do not report results beyond three years, which is accurate. However, RTOG 0915 and RTOG 0236 have presented 5-year outcomes, with LC rate of 89.4-93.2%, and RTOG 0618 presented a 4-year LC of 96%. The local control rates observed in these RTOG studies are comparable to those reported in this study. I suggest discussing the results within the context of the outcomes of the RTOG studies.

7. 24 out of 32 patients died due to "other disease" (page 15, line 248-9). The high rate of competing non-cancer mortality can diminish the incidence of cancer-related events, which has been an issue in comparing SBRT studies. It is essential to discuss how competing non-cancer mortality affects the assessment of treatment outcomes.

8. In the conclusion, it is inappropriate to suggest that combined treatment improved survival, as this is a single-arm cohort study without a comparison group. Furthermore, stating that the rates of adverse events were acceptable significantly understates the 38% incidence of pneumothorax requiring chest drainage and possibly hospitalisation. This incidence does not compare favorably with the low occurrence of adverse events observed following SBRT alone. Conclusions should take into account the significant limitations of the study.

9. Comparing Kaplan-Meier graphs for OS and DSS (Fig. 4 and 6), the number of patients at risk is nearly identical in both graphs. Given that 75% of the deaths were due to non-cancer causes, there should be a notable difference in the numbers at risk. The statistical analyses were performed using Excel, which I understand does not have a built-in Kaplan-Meier graph function and thus requires a third-party add-in or some manual steps. I recommend verifying the results using alternative statistical software.

Minor issues

1. T1-2 tumour with tumour size > 2cm as per 8th edition will be T1c-T2. I suggest using T1c-T2 instead of a lengthier description.

2. Abstract: The last line in the Methods (page 3, line 50), “Median follow-up duration was 73 months”, is a repetition since the median follow-up is mentioned in the first paragraph (line 42).

3. Table 1: The breakdown of T stage % adds up to only 91%.

4. Tables 3 and 4: the last column representing the cause of mortality is labelled “Prognosis, " which refers to the chance of an event happening. I suggest labelling it mortality and mentioning whether it is due to cancer or other causes for each patient.

5. The terms pneumonia and pneumonitis are used interchangeably to describe radiation pneumonitis. I suggest using radiation pneumonitis for consistency. This applies to tables 4 and 7 as well.

6. I suggest labelling Fig. 2 as a Consort flow diagram instead of “ patient algorithm”. Use terms such as assessed for eligibility, excluded, included in analysis, etc, to label each box clearly.

7. I suggest labelling the last subheading as Discussion rather than Comments.

8. There is use of the term “other disease” in several places, which I presume refers to a disease other than cancer. However, I am unsure if it excludes second cancers. This needs clarification.

Reviewer #3: Line 100-102: Sentence not clear.

Line 131-136: I don’t see any information on setting.

Line 135: Are those numbers in the brackets dates or what?

Line 146: What cut off was classified as old age?

Line 148-149: Did you have to test before determining if tumor was outside the reach of the probe?

Be consistent with the use of your terms. Mixing terms like local recurrence, survival, overall survival, local control makes it confusing.

Line 222: P≤0.15?? Why?

Line 233-236: Were there multiple reasons for participation? Numbers seem to exceed 66 by a lot.

Line 239 and 241: IQR should be checked.

Line 243: You followed some participants for 3 months. What happened if they were not lost to follow-up? Were they deceased?

Line 268: Is it the traditional p-value? Seems unusually large. Any justification for that?

Line 269: You set p value at 0.05, so why do you consider 0.09 significant

Line 282: What do you mean by comment?

You could explore the influence of patient’s prognosis on the outcomes. Poor prognosis can most likely after survival.

You need to provide a more thorough discussion for your findings.

Line 320-324: Don’t just state the limitations of the studies but proceed to provide suggestions for future steps.

Work on conclusion and make it better.

Reviewer #4: This study investigates a relatively uncommon treatment strategy—SBRT followed by cryoablation—for early-stage NSCLC patients who are ineligible for surgery. Reporting favorable 5-year outcomes in this population is noteworthy and clinically interesting, especially given the limited data available for this specific combination approach. The study’s strength lies in its long follow-up period and the effort to explore a minimally invasive option for patients who are not surgical candidates.

However, despite the clinical relevance and the encouraging outcomes, the lack of a control group, unclear patient selection criteria, and potential selection bias make it difficult to draw strong conclusions about the superiority or added benefit of the combined approach. The conclusions, as currently written, overstate the findings beyond what the data can support. Meaningful revisions to the manuscript are necessary to improve clarity, transparency, and scientific robustness.

- Justification for SBRT and Cryoablation Strategy

The rationale behind applying cryoablation after SBRT is not adequately justified. Authors suggest that OS might improve by preventing distant recurrence or eradicating residual tumor, but this hypothesis requires more mechanistic and clinical justification.

It is recommended that the Introduction and Discussion sections provide a more detailed and logically structured explanation of why this combination strategy was pursued. Ideally, this justification should be supported by relevant preclinical or clinical literature. Including references that explain potential synergistic effects or biological rationale would strengthen the scientific foundation of the study design.

- Patient Selection and Treatment Allocation

It is unclear how the treatment modality was determined among 333 patients. What were the specific indications for: Surgery, SBRT alone, Cryoablation alone, or SBRT followed by cryoablation. This ambiguity severely limits the interpretability of the study results and introduces significant selection bias.

For example, cryoablation could only be performed for “tumors located within the reach of the cryoprobe,” suggesting technical feasibility was a selection factor. This should be explicitly acknowledged and addressed as a source of bias.

- Need for Comparator Group

The study would benefit greatly from a matched SBRT-alone cohort to determine whether the additional cryoablation offers real benefit in terms of OS, LC, or adverse events. Without such a comparison, the claim that combination therapy is superior to SBRT alone is not supportable.

A statement like “SBRT followed by cryoablation improves survival compared to SBRT alone” (e.g., lines 60–63, 328–330) is speculative and not supported by direct comparison in this study. The study lacks a control group of patients receiving SBRT alone. Thus, the conclusion should be softened (e.g., “SBRT followed by cryoablation showed favorable outcomes…”).

- Patterns of Recurrence

It would significantly strengthen the paper to report patterns of recurrence (local, regional, or distant) and provide detail on subsequent treatments received, especially in recurrent cases.

- Toxicity Assessment

The distinction between bronchopneumonia (BP) and radiation pneumonitis (RP) is not clearly explained. What were the diagnostic criteria used to differentiate the two?

CTCAE version 4.03 is stated to be used in the main text, but Reference 17 refers to CTCAE v3.0, which should be corrected.

- Overuse of Figures and Tables

Consider combining Figures 3–6 into a single multipanel survival figure (e.g., Kaplan-Meier curves).

Tables 3–5 may be more appropriate as Supplementary Tables, as they do not add substantial value to the main narrative.

- Outdated References

Many references are more than a decade old. Recent data on SBRT, cryoablation, and combination therapies should be incorporated to better contextualize the study findings.

Reviewer #5: The manuscript presents a well-designed retrospective cohort study on the long-term outcomes of combined SBRT and cryoablation for cT1-2N0M0 NSCLC with tumor size ≥2 cm. The median follow-up of 73 months and focus on a clinically relevant population – patients ineligible or unwilling to undergo surgery – enhance the significance of this work. The results are clearly reported and show promising local control and overall survival outcomes.

However, several areas require clarification, elaboration, and improvement to strengthen the manuscript's scientific rigor and clinical impact. Please refer to the attached documents of suggested revision.

Reviewers' comments:

Reviewer's Responses to Questions

**Comments to the Author**

1. If the authors have adequately addressed your comments raised in a previous round of review and you feel that this manuscript is now acceptable for publication, you may indicate that here to bypass the “Comments to the Author” section, enter your conflict of interest statement in the “Confidential to Editor” section, and submit your "Accept" recommendation.

Reviewer #2: (No Response)

Reviewer #3: (No Response)

Reviewer #5: (No Response)

2. Is the manuscript technically sound, and do the data support the conclusions?

Reviewer #2: Yes

Reviewer #3: Yes

Reviewer #5: Partly

3. Has the statistical analysis been performed appropriately and rigorously? 

Reviewer #2: Yes

Reviewer #3: No

Reviewer #5: Yes

4. Have the authors made all data underlying the findings in their manuscript fully available?

Reviewer #2: Yes

Reviewer #3: No

Reviewer #5: Yes

5. Is the manuscript presented in an intelligible fashion and written in standard English?

Reviewer #2: Yes

Reviewer #3: Yes

Reviewer #5: No

6. Review Comments to the Author

Reviewer #2: A point-by-point response to all the reviewer comments has not been provided. Severe issues I raised have not been addressed and remain unresolved.

Reviewer #3: Other commnets seem to have been addressed. I do not see any of my comments provided earlier addressed. 

Reviewer #5: Thank you for the opportunity to review this revised manuscript entitled “Long-term outcomes of stereotactic body radiation therapy followed by cryoablation using liquid nitrogen for stage I non-small cell lung cancer with tumor size ≥2 cm.”

I commend the authors for their thorough revision in addressing the reviewers’ previous comments and for the scientific merit of this study, which explores an important combination therapy approach for early-stage NSCLC. The manuscript demonstrates solid scientific content, with clear objectives and appropriate methodology.

However, I note that the manuscript does not yet fully meet journal writing standards in terms of its scientific writing presentation. Specifically:

• Tables and figures are cited within the text but lack adequate integration and explanation. Standard journal writing requires that each table and figure is not only referenced but also introduced with context, summarised with key findings, and interpreted within the Results and Discussion. Currently, many citations simply direct the reader to the figure or table without providing an explanation of its relevance or implications.

• Despite native revision, the language and grammar remain awkward in several sections, with phrasing that may impede clarity for an international readership. Further professional language editing is required to ensure fluency, precision, and consistency.

• The results presentation tends to read as raw data reporting rather than a scientific narrative, which is expected in journal publications to guide readers through the study findings and their implications systematically.

I recommend major language editing and revision of the Results and Discussion sections, with attention to:

• Introducing each table and figure with purpose.

• Explaining what each demonstrates.

• Interpreting how each finding addresses the study objectives and contributes to the field.

Overall, I believe that with these revisions, the manuscript will reach the presentation standards required for publication and effectively communicate its important findings to the wider oncology and radiology community.

**Editor Notes: ** Please address two rounds of concerns of Reviewer #5 at the same time (previous and current questions).

7. PLOS authors have the option to publish the peer review history of their article (what does this mean? ). If published, this will include your full peer review and any attached files.

**Do you want your identity to be public for this peer review?** For information about this choice, including consent withdrawal, please see our Privacy Policy .

Reviewer #2: No

Reviewer #3: No

Reviewer #5: No

---

## [Decision Letter · Decision Letter 2]

12 Aug 2025

PONE-D-25-05871R2Long-term outcomes of stereotactic body radiation therapy followed by cryoablation using liquid nitrogen for stage I non-small cell lung cancer with tumor size ≥2 cmPLOS ONE

Dear Dr. Nomori,

Thank you for submitting your manuscript to PLOS ONE. After careful consideration, we feel that it has merit but does not fully meet PLOS ONE’s publication criteria as it currently stands. Therefore, we invite you to submit a revised version of the manuscript that addresses the points raised during the review process.

We look forward to receiving your revised manuscript.

Kind regards,

Xing-Xiong An, M.D.

Academic Editor

PLOS ONE

Journal Requirements:

Reviewers' comments:

Reviewer's Responses to Questions

**Comments to the Author**

1. If the authors have adequately addressed your comments raised in a previous round of review and you feel that this manuscript is now acceptable for publication, you may indicate that here to bypass the “Comments to the Author” section, enter your conflict of interest statement in the “Confidential to Editor” section, and submit your "Accept" recommendation.

Reviewer #3: All comments have been addressed

Reviewer #5: All comments have been addressed

2. Is the manuscript technically sound, and do the data support the conclusions?

Reviewer #3: Yes

Reviewer #5: Yes

3. Has the statistical analysis been performed appropriately and rigorously? 

Reviewer #3: Yes

Reviewer #5: Yes

4. Have the authors made all data underlying the findings in their manuscript fully available?

Reviewer #3: No

Reviewer #5: Yes

5. Is the manuscript presented in an intelligible fashion and written in standard English?

Reviewer #3: Yes

Reviewer #5: No

6. Review Comments to the Author

Reviewer #3: (No Response)

Reviewer #5: Thank you for your submission and for the thoughtful revisions made in response to prior feedback. Your study presents a novel and clinically relevant approach by combining stereotactic body radiation therapy (SBRT) with cryoablation for stage I non-small cell lung cancer (NSCLC) with tumors ≥2 cm. The long-term follow-up and outcome data are valuable and contribute meaningfully to the literature, particularly in addressing the limitations of SBRT or ablation alone in this patient population.

The manuscript demonstrates scientific merit and methodological rigor. The rationale for the combination therapy is well-articulated, and the authors have made commendable efforts to clarify the study design, patient selection, and treatment protocols.

However, I must respectfully note that the overall language and writing style remain below the standard typically expected for publication in a scientific journal. Despite revisions, the manuscript still contains awkward phrasing, grammatical inconsistencies, and structural issues that affect clarity and readability. These issues may hinder the accessibility and impact of your findings for an international readership.

I encourage the authors to consider a thorough professional language edit to improve the manuscript’s fluency, coherence, and scientific tone. Doing so would better reflect the quality of the research and enhance its suitability for publication.

7. PLOS authors have the option to publish the peer review history of their article (what does this mean? ). If published, this will include your full peer review and any attached files.

**Do you want your identity to be public for this peer review?** For information about this choice, including consent withdrawal, please see our Privacy Policy .

Reviewer #3: No

Reviewer #5: No

---

## [Author Response · Author response to Decision Letter 3]

1 Sep 2025

Response to the comments of the reviewer #5.

Thank you for your advice.

I have revised the paper according to the professional editing, which is the 3rd time.

---

## [Editor Report · Decision Letter 3]

8 Sep 2025

Long-term outcomes of combination therapy with stereotactic body radiation therapy plus cryoablation using liquid nitrogen for stage I non-small cell lung cancer with tumors ≥2 cm

PONE-D-25-05871R3

Dear Dr. Nomori,

We’re pleased to inform you that your manuscript has been judged scientifically suitable for publication and will be formally accepted for publication once it meets all outstanding technical requirements.

Kind regards,

Xing-Xiong An, M.D.

Academic Editor

PLOS ONE

Additional Editor Comments (optional):

Thanks for the authors' efforts to comprehensively improve your manuscript according to editor's and reviewers' comments. I am pleased to inform you that your paper can be accepted for publication now. Thanks for the chance to assess your work. Additionally, many thanks for all the reviewers' precious inputs.
---

## [Editor Report · Acceptance letter]

PONE-D-25-05871R3

PLOS ONE

Dear Dr. Nomori,

I'm pleased to inform you that your manuscript has been deemed suitable for publication in PLOS ONE. Congratulations! Your manuscript is now being handed over to our production team.

Kind regards,

on behalf of

Dr. Xing-Xiong An

Academic Editor

PLOS ONE